# Exome Sequencing for the Diagnostics of Osteogenesis Imperfecta in Six Russian Patients

Yulia S. Koshevaya [1], Mariia E. Turkunova [1,2], Anastasia O. Vechkasova [1], Elena A. Serebryakova [1], Maxim Yu. Donnikov [3], Svyatoslav I. Papanov [4], Alexander N. Chernov [5,6,*], Lev N. Kolbasin [3,4], Lyudmila V. Kovalenko [3], Andrey S. Glotov [6] and Oleg S. Glotov [6,7,*]

1    Saint-Petersburg State Medical Diagnostic Center (Genetic Medical Center), 194044 Saint Petersburg, Russia; yulkoshevaya@gmail.com (Y.S.K.); mariia.turkunova@gmail.com (M.E.T.); vechkasova.nastia@mail.ru (A.O.V.); el.a.serebryakova@mail.ru (E.A.S.)

2    Federal State Budget Institution of Higher Education "North-Western State Medical University named after I.I Mechnikov", Ministry of Public Health of the Russian Federation, 191015 Saint Petersburg, Russia

3    Department of Children's Diseases, Medical Institute of Surgut State University, 628400 Surgut, Russia; donnikov@gmail.com (M.Y.D.); kollev@yandex.ru (L.N.K.); kovalenko_lv@surgu.ru (L.V.K.)

4    Surgut Disctrict Clinical Center of Maternity and Childhood Health Care, 628400 Surgut, Russia; spapanov@list.ru

5    Department of General Pathology and Pathological Physiology, Institute of Experimental Medicine, 197376 Saint Petersburg, Russia

6    Department of Genomic Medicine, D. O. Ott Research Institute of Obstetrics, Gynecology and Reproductology, 199034 Saint Petersburg, Russia; anglotov@mail.ru

7    Department of Experimental Medical Virology, Molecular Genetics and Biobanking of Virological and Molecular Genetic Methods of Diagnostics of Children's Scientific and Clinical Center for Infectious Diseases of the Federal Medical and Biological Agency, 197022 Saint Petersburg, Russia

\*    Correspondence: alexanderchernov1981@gmail.com (A.N.C.); olglotov@mail.ru (O.S.G.)

**Abstract:** Osteogenesis imperfecta (OI) is a group of inherited disorders of connective tissue that cause significant deformities and fragility in bones. Most cases of OI are associated with pathogenic variants in collagen type I genes and are characterized by pronounced polymorphisms in clinical manifestations and the absence of clear phenotype–genotype correlation. The objective of this study was to conduct a comprehensive molecular–genetic and clinical analysis to verify the diagnosis of OI in six Russian patients with genetic variants in the *COL1A1* and *COL1A2* genes. Clinical and laboratory data were obtained from six OI patients who were observed at the Medical Genetics Center in Saint Petersburg from 2016 to 2023. Next-generation sequencing on MGISEQ G400 (MGI, China) was used for DNA analysis. The GATK bioinformatic software (version 4.5.0.0) was used for variant calling and hard filtering. Genetic variants were verified by the direct automatic sequencing of PCR products using the ABI 3500X sequencer. We identified six genetic variants, as follows pathogenic c.3505G>A (p. Gly1169Ser), c.769G>A (p.Gly257Arg), VUS c.4123G>A (p.Ala1375Thr), and c.4114A>T (p.Asn1372Tyr) in *COL1A1*; and likely pathogenic c.2035G>A (p.Gly679Ser) and c.739-2A>T in *COL1A2*. In addition, clinical cases are presented due to the presence of the c.4114A>T variant in the *COL1A2* gene. Molecular genetics is essential for determining different OI types due to the high similarity across various types of the disease and the failure of unambiguous diagnosis based on clinical manifestations alone. Considering the variable approaches to OI classification, an integrated strategy is required for optimal patient management.

**Keywords:** osteogenesis imperfecta; *COL1A1*; *COL1A2*; multiple fractures; molecular and genetic diagnostics

## 1. Introduction

Osteogenesis imperfecta (OI) is a group of inherited disorders of connective tissue that cause significant deformities and fragility in bones, leading to minor traumatic or atypical fractures, detected in the prenatal and postnatal periods or in early childhood [1].

The prevalence of OI is about 1:10,000–20,000 newborns and depends on the form of the disease (severe forms have less prevalence than mild forms) and the region where the study has been carried out [1,2]. In Europe, the prevalence of OI is 1:15,000 [3]. In North America, the prevalence of OI is 8:100,000 [4].

The clinical OI manifestations include tooth anomalies, progressive hearing loss, blue sclera, joint hypermobility, and, less often, cardiovascular and respiratory insufficiency and muscle weakness [1,5]. The reason for the higher fracture frequency in OI patients is osteoporosis, which results from a progressive decrease in bone mass [6]. Respiratory complications occur as a result of spine or chest deformities due to decreased mobility and elasticity [7]. Dental anomalies such as brittle, missing, or ectopic teeth or malocclusion are common findings in OI and are associated with severe disease. Imperfect dentinogenesis causes abnormally yellow and translucent coloration of teeth, premature tooth wear, and breakage. The radiographic assessment of dental anomalies shows shorter roots with coronaradicular abrasions [8].

Most cases of OI are caused by genetic modifications in collagen type I genes, which are encoded by the *COL1A1* and *COL1A2* genes [9]. The *COL1A1* null alleles lead to haploinsufficiency, and missense variants lead, respectively, to mild or severe signs or death [10]. These mutations generally arise de novo [11]. The screening of the *COL1A1* and *COL1A2* genes covers the most common molecular causes of OI, but only a minority of the types.

Thus, OI diagnostics should be based on clinical data and molecular genetic testing for identification of the OI type and perform differential diagnosis with other connective tissue dysplasias (hypophosphatasia, Ehlers–Danlos syndrome, etc.). For OI diagnostics, ultrasound, computerized tomography (CT) scanning, magnetic resonance angiography (MRA) screening, and PCR analysis are usually used [2]. Recently, whole-exome sequencing (WES) has begun to be used to diagnose OI [12]. Using molecular genetic testing and WES allows us to assess OI severity and choose the most optimal treatment strategy.

The objective of this study is to use WES to carry out a comprehensive molecular–genetic and clinical analysis to verify OI diagnosis in Russian patients.

## 2. Materials and Methods

Our paper presents clinical and laboratory evidence obtained from six OI patients. Patients were observed at the Medical Genetics Center from 2016 to 2023. All patients signed a written informed consent to use their data for research, including the present publication.

### 2.1. Samples Preparations

Blood samples from all patients and several patients' family members were collected. We collected samples from parents of patients #1, 3, 5, and 6. Unfortunately, we had no opportunity to investigate parents in family cases of OI (#4 and 2). Blood samples were stored using complex laboratory facilities for large-scale studies (#3076082, "Human Reproductive Health"). For all blood samples, DNA was isolated using the phenol extraction method. Quantus FluorometerTM and QuantiFluor R dsDNA System (Promega Corporation, Madison, WI, USA) were used to determine DNA concentration. DNA electrophoresis in a 0.6% agarose gel in sodium borate (SB) buffer was used to assess DNA integrity.

### 2.2. Whole-Exome Sequencing

All patients underwent WES to identify molecular genetic causes of OI. The MGISEQ G400 (MGI, Shenzhen, China) integrated next-generation sequencing instrument was used for DNA analysis based on $2 \times 100$ bp pair-end reads with at least $70\times$ (79–129$\times$) mean target coverage.

Variants were also verified by the direct automatic sequencing of PCR products using the ABI 3500X sequencer (Applied Biosystems™, Pleasanton, CA, USA). WES was carried out after the ethical committee's approval in 2022. Targeted capture technique was used to selectively sequence complete coding regions of the exome. The standard HGVS nomen-

clature was used to name the identified variants (https://mutalyzer.nl/ version 2.0.25, accessed on 10 March 2024). The fragmented DNA was then converted into DNA libraries using KAPA Hyper Prep Kit (Roche, Basel, Switzerland) in combination with MGIEasy DNA Adapters-96 (MGI, Shenzhen, China). Exome-enrichment of DNA libraries was performed using Hyper Cap Target Enrichment kit and KAPA Hyper Exome Probes set (Roche, Basel, Switzerland), according to the manufacturer's protocol [13], with the following modifications: 1 μL of Block3 and 10 μL of Block4 reagents from the MGIEasy Exome Capture Accessory kit were added to the hybridization mix instead of KAPA Universal Enhancing Oligos, and the final library amplification was performed using MGI PCR Primer Mix. Library quantitation was performed using Quantus Fluorometer with QuantiFluor® dsDNA System kit (Promega, Madison, WI, USA) [14]. A High-Sensitivity DNA assay with gel electrophoresis using the 2100 Bioanalyzer System (Agilent Technologies, Santa Clara, CA, USA) was performed for DNA sizing accuracy and quality control (between 300 and 400 bp) [15]. Paired-end reads not shorter than 100 bp were generated for each sample. The method has limitations and does not include the study of non-coding regions. The method is not intended for assessing the level of methylation, identifying chromosomal rearrangements (translocations, insertions, deletions, duplications, and inversions of more than 10 nucleotide pairs in the coding regions of genes), detecting aneuploidy and polyploidy, identifying mutations in the state of mosaicism, or analyzing variations in repeat length (including expansion triplets).

*2.3. Bioinformatic Data Analysis and Variant Calling in Patient Exomes*

Each exome sample was aligned onto a GRCh38.p13 reference genome assembly provided in the Genome Analysis ToolKit (GATK) (Cambridge, MA, USA) [16] bundle using the BWA MEM read aligner [17]. GATK software (version 4.5.0.0) was used for variant calling and hard filtering [16] in compliance with the BROAD Institute guidelines [18]. Transcript-aware variant annotation, defining every variant as the sum of transcripts reported in the RefSeq and Locus Reference Genomic databases, utilized pathogenicity prediction tools (SIFT, PolyPhen2, PROVEAN, fathmm-MKL) and computational methods to estimate evolutionary changes and conservation scores for all positions (GERP, PhyloP). To estimate the population frequencies of the identified SNPs, samples from the Exome Aggregation Consortium, Genome Aggregation Database, Exome Variant Server, and 1000 Genomes Project projects were used. Clinical relevance of the identified variants was assessed using the dbSNP, ClinVar, OMIM, HGMD, DMDM, and LOVD databases and publications by fellow investigators [19]. Variants were verified by the direct automatic sequencing of PCR products using the ABI 3500X sequencer. The NM_000088.4 reference sequence (RefSeq database) provided a stable reference for variant annotation. The HGVS nomenclature was used to describe sequence variants [20–22].

*2.4. Statistical Analysis*

Phenotypic severity was assessed using the $\chi^2$ Cochrane–Armitage test for trend and stepwise Bonferroni correction for multiple comparisons (Bonferroni–Hochberg procedure) [23]. Values of $p < 0.05$ were treated as statistically significant.

**3. Results**

The clinical and genetic data of Russian OI patients are presented in Tables 1 and 2.

**Table 1.** Genetic data of OI patients.

| Patient No. | Gene, Coverage | Gene Variant | Pathogenicity; Pathogenic Criteria | OI Type | Gender |
|---|---|---|---|---|---|
| 1 | *COL1A1*, 90× | c.4123G>A (p.Ala1375Thr) ° | VUS; PM, PP3, PP4 | Type III | Female |
| 2 | *COL1A1*, 96× | c.3505G>A (p.Gly1169Ser) | Pathogenic; PS4, PM1, PM2, PP1, PP2, PP3, PP5 | Type III (family history) | Female |
| 3 | *COL1A1*, 79× | c.769G>A (p.Gly257Arg) ° | Pathogenic; PS4, PM2, PM5, PP2, PP3, PP5 | Type III | Female |
| 4 | *COL1A1*, 85× | c.4114A>T (p.Asn1372Tyr) * | VUS; PM2, PP3, PP4 | Type III (family history) | Male |
| 5 | *COL1A2*, 129× | c.2035G>A (p.Gly679Ser) ° | Likely pathogenic; PM2, PM5, PP3, PP4, PP5 | Type III | Male |
| 6 | *COL1A2*, 82× | c.739-2A>T °* | Likely pathogenic; PVS1, PM2 | Type III | Female |

Note: * Variant we are describing for the first time; ° de novo variants; PM: pathogenic moderate; PP: pathogenic supporting; PS: pathogenic strong; PVS: pathogenic very strong; VUS: variant of uncertain value. These pathogenicity criteria are derived from the American College of Medical Genetics and Genomics recommendations [24].

**Table 2.** Clinical data of OI patients.

| Patient No. | Clinical Manifestations | Age of Onset | Phenotype | Fractures | Biochemistry | Other Medication |
|---|---|---|---|---|---|---|
| 1 | By 1 month of age: 6 fractures (low trauma or in atypical locations) | After birth | By 7 years and 1 month: short stature (−2.57 sd), overweight (−2.5 sd BMI), prominent frontal eminence, varus deformity of the lower extremities | After-birth fractures: left humerus, left femur and fibula, both radial bones. Compression fractures of the Th11-L2 vertebrae, ribs 3 to 10 on the right. Fractures by 7 years and 1 month: at 6 and 10 months, fractures of the right leg and left arm, respectively | By 7 years and 1 month: Ca, P, and AP in the reference range. High level of 25-(OH)VitD3 in the anamnesis (overdose of cholecalciferol) | Calcium and cholecalciferol |
| 2 | By 1 month of age: 4 fractures (low trauma or in atypical locations) | In utero | By 3 months: blue sclera, varus deformity of the lower extremities and ribs | In utero fractures: left femur. Fractures by 3 months: both femurs and both tibias | By 3 months: Ca and P in the reference range. High level of AP | Calcium and cholecalciferol |
| 3 | By 2 years of age: 12 fractures (low trauma or in atypical locations) | 1 month | By 3 years and 4 months: height (−1.65 sd), weight (−1.25 sd BMI). Scoliosis of the thoracic and lumbar spine, asymmetrical leg length, deformation of the right hip. Varus deformity of the humerus, legs, and thighs | Fractures at 1 month: both femurs. By 3 years and 4 months: multiple fractures of long tubular bones; compression fractures of Th5-Th9, 12th, and L2 vertebrae; fracture of the left clavicle (together more than 19th) | By 3 years and 4 months: low Ca and AP, high 25-(OH)VitD3 level in the anamnesis (overdose of cholecalciferol) | Calcium and cholecalciferol |
| 4 | By 1 month of age: 7 fractures (low trauma or in atypical locations) | In utero | By 3 months: short stature (−3.9 sd), low weight (−2.53 sd IBM), prominent frontal eminence | In utero fractures: left femur, ribs, right humerus. Fractures by 3 months: ribs 5, 6, 7, 8, and 9; both radius, both femurs, right femur in 2 positions, right humerus in 2 positions | By 3 months: Ca, P, and AP in the reference range | Calcium and cholecalciferol |

**Table 2.** *Cont.*

| Patient No. | Clinical Manifestations | Age of Onset | Phenotype | Fractures | Biochemistry | Other Medication |
|---|---|---|---|---|---|---|
| 5 | By 13 years of age: 9 fractures (low trauma or in atypical locations) | 12 months | By 13 years: normal height (0.47 with BMI), overweight (+2.5 with BMI). Blue sclera, asymmetrical leg length, disproportionate physique (short torso), hypodontia, joint hypermobility, signs of connective tissue dysplasia | Fracture at 1 year: left tibia Fractures by 13 years: at 3 years—temporal bone and compression fracture of vertebrae Th6 and 8; at 10 years—bone of the left forearm; at 11 years—right tibia; at 12 years—left radius and right femoral neck; at 13 years—left tibia | By 13 years: Ca, P, and AP in the reference range, low 25-(OH)VitD3 and parathyroid hormone levels in the anamnesis | Calcium and cholecalciferol |
| 6 | By 2 years of age: 3 fractures (low trauma or in atypical locations) | 1 month | | Fracture at 1 month: left forearm Fractures by 2 years: fracture of both lower extremities | By 2 years: Ca, P, and AP in the reference range | Cholecalciferol |

Note: AP—alkaline phosphatase.

In some cases, skeletal deformations in OI are similar to those in other forms of skeletal dysplasia, which can manifest themselves in utero, as was detected in some of our patients. In addition, our patients had signs of connective tissue dysplasia, including muscle hypotonia, scoliosis, and joint hypermobility, which can occur in various hereditary connective tissue disorders. Although OI is usually not associated with disorders of phosphorus–calcium metabolism, some of our patients had abnormal levels of alkaline phosphatase and parathyroid hormone. This was not permanent, but we had to perform a differential diagnosis of hypophosphatasia and other ricket-like diseases in which multiple fractures and bone deformations are also possible.

The obtained evidence and clinical manifestations, as well as the identified pathogenic SNPs in the *COL1A1* and *COL1A2* genes, allowed us to diagnose OI types in our patients. The mutations in four patients were de novo (Table 1). The parents of patients #1, 3, 5, and 6 had no fractures in their anamnesis and no mutations in the *COL1A1* and *COL1A2* genes, respectively.

*COL1A1* gene mutations c.769G>A (p.Gly257Arg) and c.3505G>A (p.Gly1169Ser) disrupt the spatial structure of collagen, ultimately leading to deteriorated mechanical properties and contributing to the development of OI [25]. Glycine is crucial for the smallest side chain of all amino acids. This allows glycine to 'squeeze' into the internal part of the helix, becoming the most critical 'puzzle' for correct collagen folding [26]. Glycine replacement disrupts the spatial structure of collagen, ultimately leading to deteriorated mechanical properties and contributing to OI development. The clinical manifestation of the c.3505G>A (p.Gly1169Ser) variant is confirmed by similar symptoms in the patient's father. The father of patient no. 2 suffered five fractures of the femur from the age of 4.5 years; after reaching adulthood, there were no fractures. No treatment or genetic analysis was performed. *COL1A1* gene mutation c.4123G>A (p.Ala1375Thr) was early detected in Japanese patients. This mutation is localized in the triple-helical domain of the *COL1A1* gene [27].

The c.2035G>A (p.Gly679Arg) mutation in the *COL1A2* gene, which replaces glycine with the larger amino acid arginine, can disrupt collagen structure, leading to abnormal protein activity. The c.739-2A>T mutation affects the canonical splice acceptor site in intron 15 of the *COL1A2* gene and appears to contribute to modifications of collagen structure and function [28]. Moreover, patients with clinically manifest OI reported another SNP in the same position (c.739-2A>G). Thus, all OI patients were diagnosed with type III based on clinical data.

The variants of c.4114A>T (p.Asn1372Tyr) have never been reported by fellow investigators and are absent in the gnomAD control sample [29]. Pathogenicity predictions in in silico programs (SIFT, MT, DANN, and MetalRL) consider these variants of uncertain value (VUSs) with the potential to disrupt protein function. The patient's phenotype is highly specific for the genetic etiology. The impact of this variant on the OI phenotype is also supported by clinical symptoms in the patient's mother. The mother of patient no. 4 was diagnosed with a severe form of OI at an early age. By the age of 32, she had lumbar scoliosis, asymmetrical leg length, osteopenia, and astigmatism. Molecular genetic testing was not performed. The overall evidence allowed us to consider such variants as being of unknown clinical significance. Since the clinical picture of the patient's father is unknown, this variant should be considered as a variant with incomplete penetrance.

## 4. Discussion

OI is a genetically heterogeneous connective tissue disorder, displaying incomplete penetrance and variable severity. The wide variety of clinical symptoms of OI causes the existence of 20 types of OI as a result of mutations in different genes and types of inheritance (Table 3) [7]. In addition to collagen, a number of genes are also involved in the pathogenesis of OI: *BMP1*, *CCDC134*, *CREB3L1*, *CRTAP*, *FKBP10*, *IFITM5*, *KDELR2*, *MBTPS2*, *MESD*, *P3H1*, *P4HB*, *PLOD2*, *PPIB*, *SEC24D*, *SERPINF1*, *SERPINH1*, *SP7*, *SPARC*, *TENT5A*, *TMEM38B*, and *WNT1* [30–32]. The complex consisting of prolyl-3-hydroxylase 1 (P3H1), cartilage-associated protein (CRTAP), and peptidyl-prolyl-cis-trans-isomerase B (PPIB) is responsible for the hydroxylation of proline-986 in the α1 chain. PPIB ensures the cis–trans isomerization of the collagen-prolyl-peptide bond and, together with FKBP10, a molecular chaperone, prevents the procollagen chains from being assembled into fibrils prematurely. Mutations in the three respective genes lead to a decrease in proline-986 hydroxylation and, thus, a delay in collagen folding [33]. Lysyl hydroxylase 2 (PLOD2) hydroxylates lysine residues in the collagen molecule. This process enables covalent crosslinking within the molecule and thus impairs the tensile strength and stability of the protein [34]. Transmembrane protein 38B (TMEM38B) is a trimeric intracellular potassium channel type B that is necessary for emptying intracellular calcium stores. A disturbed intracellular calcium release leads to an incorrect regulation of collagen modification by various enzymes in the ER. This results in ER stress and reduced collagen secretion [35]. The dysfunction of membrane-bound transcription factor peptidase site 2 (MBTPS2) leads to reduced hydroxylation of a lysine residue, disturbed collagen crosslinking, and reduced collagen secretion [36]. Secreted protein acidic and rich in cysteine (SPARC) can serve as a molecular chaperone during collagen biosynthesis. In the extracellular space, SPARC mediates extracellular matrix–cell interactions and promotes mineralization of the extracellular matrix by binding to collagen and hydroxyapatite [37]. The *PLS3* gene codes for the cytoskeletal protein plastin-3. Mutations in *PLS3* lead to reduced trabecular thickness with normal expression and modification of collagen [33]. Mutations in the *WNT1* gene trigger altered signal transduction and restricted expression of osteoblast-specific genes regulating bone cell homeostasis. The patients with *WNT1* mutations show reduced bone remodeling, indicating an imbalance between bone formation and resorption [38].

**Table 3.** Genes, types, and clinical signs of osteogenesis imperfecta [4,7,9].

| Type | Gene | Inheritance | Gene Product | Clinical Presentation |
|------|------|-------------|--------------|-----------------------|
| I | *COL1A1* | AD | Collagen type I, α1-chain | - mild severity<br>- normal or near-normal growth velocity and height<br>- blue-gray sclerae<br>- conductive hearing loss<br>- minor long tubular bone deformity |
| II | *COL1A1, COL1A2* | AD | Collagen type I, α1- and α2-chain | - high risk of perinatal death<br>- occasional intra-uterine bone fracturing<br>- intra-uterine rib fractures (radiographically beaded ribs and crumpled accordion-like long tubular bones)<br>- deficiency of ossification of facial and skull bones |
| III | *COL1A1, COL1A2* | AD | Collagen type I, α1- and α2-chain | - newborn or infant presentation of severe progressive bone deformity<br>- progressive kyphoscoliosis<br>- stunted growth |
| IV | *COL1A1, COL1A2* | AD | Collagen type I, α1- and α2-chain | - recurrent bone fractures<br>- long tubular bone deformities of variable severity<br>- osteoporosis<br>- normal sclerae<br>- posterior fossa compression syndrome due to basilar impression with elevation of the floor of the posterior cranial fossa |
| V | *IFITM5* | AD | Interferon-induced transmembrane protein 5 | - moderately severe<br>- calcification in interosseous membranes of forearm and increased propensity to develop hyperplastic callus |
| VI | *SERPINF1* | AR | Serpin peptidase inhibitor | - frequent bone fractures<br>- joint hypermobility<br>- stunted growth<br>- blue sclerae<br>- accumulation of unmineralized osteoid (characterized by osteomalacia) |
| VII | *CRTAP* | AR | Cartilage protein | - multiple bone fractures<br>- rhizomelia<br>- micromelia |
| VIII | *LEPRE1* | AR | Prolyl 3-hydroxylase 1 | - multiple bone fractures<br>- bulbous metaphyseal expansion of bones |
| IX | *PPIB* | AR | Peptidyl-prolyl isomerase B | - multiple bone fractures<br>- short and flexed hips; curved hips with bended anterior tibia |
| X | *SERPINF1* | AR | Serpin peptidase inhibitor | - multiple bone fractures<br>- imperfect dentinogenesis<br>- mid-face hypoplasia |
| XI | *FKBP10* | AR | FKBP10 peptidyl-prolyl cis–trans isomerase | - multiple bone fractures<br>- hyperosteoidosis<br>- joint contractures<br>- bone microscopy: bone tissue has a distinct 'fish-scale' pattern |

**Table 3.** *Cont.*

| Type | Gene | Inheritance | Gene Product | Clinical Presentation |
|---|---|---|---|---|
| XII | *SP7* | AR | Zinc finger-containing transcription factor | - multiple bone fractures<br>- progressive hearing loss<br>- stunted growth<br>- hyperextensibility of interphalangeal joints |
| XII | *BMP1* | AR | Bone morphogenetic protein 1 | - bone fractures<br>- long tubular bone deformity<br>- joint hypermobility |
| XIV | *TMEM38B* | AR | Transmembrane protein 38B | - bone fractures<br>- stunted growth<br>- pseudoarthrosis<br>- deformity of the femur and tibia |
| XV | *WNT1* | AR | Oncogenic signaling protein | - multiple bone fractures<br>- long tubular bone deformity<br>- brain defects |
| XVI | *CREB3L1* | AR | cAMP-responsive element-binding protein 3 | - multiple bone fractures (may occur in utero)<br>- beaded ribs<br>- hearing loss<br>- tooth agenesis<br>- bone calluses<br>- shorter arms and legs |
| XVII | *SPARC* | AR | Protein rich in cysteine | - bone fractures<br>- joint hypermobility<br>- psychomotor and speech delay<br>- muscle weakness |
| XVIII | *FAM64A/TENT5A* | AR | Mitotic regulator protein/Terminal nucleotidyltransferase | - multiple bone fractures<br>- psychomotor and speech delay<br>- dysmorphic facial features |
| XIX | *MBTPS2* | XLR | Membrane-bound transcription factor protease, Site 2 | - multiple bone fractures<br>- stunted growth<br>- calcifications in the epiphyseal region of bones<br>- generalized osteopenia |
| Unclassified | *PLOD2* | AR | Procollagen-lysine 2-oxoglutarate 5-dioxygenase | - bone fractures<br>- joint contractures |
| Unclassified | *PLS3* | XLR | Plastin 3 | - bone fractures<br>- osteoporosis |

Note: AD—autosomal dominant; AR—autosomal recessive; XLR—X-linked recessive.

The main diagnostic criteria include a high incidence of fractures, osteoporosis, and stunted growth. All the clinical manifestations are in correlation with the OI type. In turn, the OI type is determined by the affected gene and its impact on the respective protein function [7].

In most cases, the correlation between genotype and clinical phenotype data is confirmed by the evidence of the presence of shifts in canonical splice sites and missense variants in the *COL1A1* and *COL1A2* genes, which induce qualitative collagen defects [38]. The collagen type I triple-helix molecule consists of two α1 chains and one α2 chain encoded by *COL1A1* and *COL1A2*, respectively. The structure of collagen is characterized by a Gly-X-Y repeat sequence, dominated by proline and hydroxyproline at the X and Y positions. Pathogenic glycine substitutions that alter collagen type I's structure are usually

associated with moderate-to-severe clinical manifestations. The replacement of a carboxyl-terminal glycine residue in *COL1A1* is associated with greater disease severity than in *COL1A2* [1]. In this research, we described the c.3505G>A (p. Gly1169Ser, rs67815019) variant in the *COL1A1* gene. Hong-Yan Liu showed that familial cases of this variant in exon 47 of the *COL1A1* gene have highly variable clinical symptoms, ranging from mild OI to severe and fatal [39]. According to the Consortium for OI, this variant is classified as pathogenic and contributes to the development of OI type I [40]. We also identified a c.769G>A (p.Gly257Arg, rs72645321) missense variant in the *COL1A1* gene in one patient. The p.Gly79 amino acid is presented in the Gly-Xaa-Yaa helical domain [41]. This variant is present in 24 publications in the ClinVar database and is identified as pathogenic (PS3, PS4, PM1, PM2, PP3). This SNP promotes the development of both OI type I and OI type III [41].

*COL1A1* null alleles lead to haploinsufficiency, usually associated with mild OI (type I). We determined a c.2035G>A (p.Gly679Ser, rs1584325552) missense variant in the *COL1A2* gene in one patient. This variant results in the replacement of a glycine residue at codon 679 of *COL1A2* within the collagen triple helix and is conserved in all mammalian species. The glycine replacement in collagen protein is indeed critical for the confined collagen protein-specific triple-helix structure and its stability. Glycine substitutions can disrupt this structure, impairing the tensile strength and stability of collagen tissues, including bones [26]. The variant is described in ClinVar as likely pathogenic and is associated with the development of Ehlers–Danlos syndrome type I and the recessive perinatal lethal form of OI [42].

On the other hand, missense variants or splicing mutations in *COL1A1* or *COL1A2* usually lead to lethal, severe, or moderate OI. These mutations generally arise de novo [7]. The *COL1A2* c.739-2A>T variant (rs72656382) was detected by us in one patient with OI. This variant is present in the ClinVar database [43]. The c.739-2A>T variant abolishes the canonical splice acceptor site of intron 15, which disrupts gene function [44]. Another variant (c.739-2A>G, p.Gly247Cys, rs1064794058) affects canonical splice sites and was detected in patients with OI type II. In silico analyses predict that this variant is damaging to the structure and function of collagen. In the Human Gene Mutation Database, the missense variants (G247R and G247D) at this residue are associated with OI [45]. Therefore, the ClinVar database interpreted this variant as a pathogenic variant [46].

Yousuke Higuchi et al., having carried out Sanger sequencing of the *COL1A1/2* and *IFITM5* genes in 96 Japanese OI probands, 44 of whom had an OI family history, found that 56.3% of patients had a mild form of OI type I, 42.7% had a moderate-to-severe form of OI types II–IV, and 1.0% had OI type V. Thus, 90% of all OI cases are caused by variants in the *COL1A1* and *COL1A2* genes. In one patient, the authors also found the variant c.4123G>A (p.Ala1375Thr) described by us in the *COL1A1* gene and classified it as likely pathogenic (PM2 PM6 PP3, PP4), with an r = 0.997 genotype–phenotype correlation coefficient [27].

Since all body tissues are involved in abnormal type I collagen production, the condition always manifests as a systemic disease. Pathogenic variants associated with modifications in the collagen structure lead to the retention of distended αI-chains in the endoplasmic reticulum (ER), causing abnormal modifications in collagen chains and fibrils. The resulting aberrant extracellular matrix is a most crucial factor responsible for bone fragility in OI. In some cases, abnormal collagen is partially retained in the ER, causing ER stress due to the stimulation of autophagy, the activation of apoptosis, and the impaired differentiation of osteoblasts. These events lead to a dysfunctional bone matrix, decreased collagen synthesis, and reduced bone density [1].

X. Lin et al. performed large cohort studies involving 560 OI patients using WES to identify OI-responsible mutations and genotype–phenotype correlations. The *COL1A1* and *COL1A2* mutations were among the most commonly reported. Other findings included the correlation between genotype and phenotype, depending on the mutation and the protein modification caused [47]. A group of investigators from Estonia, Vietnam, and Ukraine showed that 82 (56.16%) patients of all the 146 examined cases revealed pathogenic de

novo variants. The de novo cases were 37.04% (10/27) among Estonian patients, 58.06% (36/62) among Ukrainians, and 63.16% (36/57) among Vietnamese [37]. Since the described variants were also found in other human populations, the sites of mutation occurrence are not related to the founder effect but rather to the mechanisms of mutation occurrence.

A molecular analysis of the *COL1A1* and *COL1A2* variants is of paramount importance to gain insight into the causes of OI. Quantitative modifications are largely associated with qualitative defects, increasingly affecting disease severity [48]. By identifying the exact mutations or their combinations, practitioners can link the genetic information with the clinical manifestations of the disease. Depending on the mutation type and the location, OI patients may present versatile clinical manifestations. This can range from mild bone deformities to more serious fractures, and it can even affect other organs [49].

## 5. Conclusions

Using comparative WES and clinical analyses, we identified molecular–genetic mutations in six Russian patients with OI and diagnosed their OI types. For the first time, we also observed a VUS (c.4114A>T, p.Asn1372Tyr) variant in the *COL1A1* gene that was associated with moderate clinical (multiple fractures) signs of type III OI in the patient. The VUS (c.4123G>A, p.Ala1375Thr) and likely pathogenic (c.2035G>A, p.Gly679Ser) variants were identified de novo in the *COL1A1* and *COL1A2* genes, respectively, and were associated with clinically moderate signs of type III OI in the patients. We also detected two pathogenic variants (c.3505G>A, p.Gly1169Ser; and c.769G>A, p.Gly257Arg) in the *COL1A1* gene that were associated with clinically strong signs of type III OI in the patients. A likely pathogenic (c.739-2A>T) de novo variant in the *COL1A2* gene in a Russian patient was associated with very strong clinical signs (multiple fractures) of type III OI. These results show that phenotype severity correlates with the affected genes, helix location, amino acid substitutions, and the resulting residue and predicted final protein product, though the complete correlation between genotype and phenotype is not entirely clear, and controversies persist due to wide phenotypic variability and the overlap with other low-bone-mass disorders. A deeper understanding of the OI genetic prerequisites and the use of WES should become a standard tool for carrying out differential diagnosis, classifying OI subtypes, helping to obtain a broad view of the genetic landscape, enabling early diagnosis, improving accuracy, assessing disease severity, and implementing efficient treatment strategies, and may become more cost-effective than gene-by-gene/panel examination. The first genes whose variants led to the development of OI were *COL1A1* and *COL1A2*, and subsequently, their number gradually increased. Currently, there are about 20 types of OI caused by mutations in various genes that were identified using exome sequencing. It is possible that, in the future, the number of types of OI may increase as new genes whose variants can lead to the development of the disease are identified. Therefore, using only targeted gene panels, we can miss patients with this pathology who would need timely and correct management. In addition, OI has clinical manifestations that are similar to other diseases characterized by increased bone fragility and/or skeletal dysplasia. In connection with the above, it is preferable to perform whole-exome sequencing in patients with OI. Therefore, WES is the most relevant method of molecular genetics for OI [50]. Considering the variable approaches to OI classification, an integrated strategy is required for optimal patient management.

**Author Contributions:** Conceptualization, O.S.G. and A.S.G.; methodology, Y.S.K., M.E.T., A.O.V. and E.A.S.; data curation, L.V.K., O.S.G., A.S.G. and A.N.C.; original draft writing, Y.S.K., M.E.T., A.O.V. and E.A.S.; review and editing, M.Y.D., S.I.P., L.N.K., O.S.G., A.S.G. and A.N.C.; visualization, Y.S.K., M.E.T., A.O.V. and E.A.S.; supervision, O.S.G. and A.S.G. All authors have read and agreed to the published version of the manuscript.

**Funding:** This study was funded by the Ministry of Science and Higher Education of the Russian Federation (project Multicenter research bioresource collection "Human Reproductive Health", contract no. 075-15-2021-1058, 28 September 2021).

**Institutional Review Board Statement:** This study was conducted in accordance with the Declaration of Helsinki (and approved by the Ethics Committee of the D. O. Ott Research Institute of Obstetrics, Gynecology, and Reproductology) (protocol code no. 117 and date of approval 19 April 2022).

**Informed Consent Statement:** Informed consent was obtained from all subjects involved in the study. Written informed consent has been obtained from the patients to publish this paper.

**Data Availability Statement:** Data is contained within the article.

**Conflicts of Interest:** The authors declare no conflicts of interest.

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
