# Peer review of "Exome Sequencing for the Diagnostics of Osteogenesis Imperfecta in Six Russian Patients"

_cimb, doi:10.3390/cimb46050252_

Round 1

Reviewer 1 Report (New Reviewer)

Comments and Suggestions for Authors

The authors in the present study WES and clinical analysis identified molecular-genetic mutations in six Russian patients with osteogenesis imperfecta. The identified several SNVs in the Collagen genes of these patients associated with the disease type and severity. I have following concern

1.    An analysis workflow is needed.

2.    Why did the author did not look at Variation analysis in other genes BMP1, CCDC134, CREB3L1, CRTAP, FKBP10, IFITM5, KDELR2, MBTPS2, MESD, P3H1, 75 P4HB, PLOD2, PPIB, SEC24D, SERPINF1, SERPINH1, SP7, SPARC, TENT5A, TMEM38B 76 and WNT1

3.    Can the author perform a network analysis of the pathogenic variants in collagen gene with the other genes associated with the disease?

4.    It would be great if authors could validate their results from data present in the publicly available data sets

5.    Please write the findings of the study briefly in the Introduction along with the rationale.

Author Response

Dear Reviewer,

Please see our answers to your questions in the attached file.

Best regards, A. Chernov, O. Glotov

Reviewer 2 Report (New Reviewer)

Comments and Suggestions for Authors

The manuscript cimb-2924611, "Exome sequencing for the diagnostics of osteogenesis imperfecta in 6 Russian patients", reports the genetic analysis by whole-exome sequencing (WES) in six Russian patients to confirm the clinical diagnosis of OI. The authors found four variants in COL1A1 and two in COL1A2. One (?) was novel and never reported in the literature.

The manuscript in its current form is not ready for publication because of several flaws that need to be adjusted. Considering its submission to the topic collection "Bioinformatics Approaches to Biomedicine", it should convey the message that due to the clinical and genetic heterogeneity of OI and the overlap with other low bone mass disorders, exome-sequencing should become a standard tool for differential diagnosis, classification of OI subtypes, helping to get a broad view of the genetic landscape, enabling early diagnosis, and may becoming more cost-effective than gene-by-gene/panel examination. This message is not adequately delivered, and therefore, it loses its efficacy. Furthermore, a profound revision, enrichment and reshaping of the content would improve the manuscript.

Here are the points, major and minor, to be addressed:

Major comments:

1.     It is unclear how many variants the authors found are novel and never reported in the literature as associated with OI. In the Abstract, lines 39, 40, "In addition, clinical cases are presented due to the previously undescribed variant c.739-2A>T in the COL1A2, c.4114A>T and c.4123G>A in the COL1A1 genes." There seem to be three variants here, while only one is in the results and Table 2. Please clarify.

2.     In the Materials and Methods:

-        lines 127-128, "Samples preparations. Blood samples from all patients and several patients' family members were collected". Please specify which members were collected. It would be helpful to have a figure representing all the six cases' pedigrees.

Lines 174-175: "Bioinformatic data analysis and variant calling in patient exomes… Variants were verified by the direct automatic sequencing of PCR products using the ABI 3500X sequencer." Please specify which family members were tested. Were all the ones who underwent exome sequencing tested? Were only probands and parents tested? It would be helpful to add this information to the previously suggested figure.

-       The authors report a Statistical Analysis paragraph, "Phenotypic severity was assessed using the χ2 Cochrane-Armitage test for trend and stepwise Bonferroni correction for multiple comparisons (Bonferroni-Hochberg procedure) [34]. Values of p <0.05 were treated as statistically significant. Statistical analysis was performed using The R Project for Statistical Computing software (v. 4.2.1)." Where are the results of this analysis?

3.     In the Results:

-        line 203, "The c.739-2A>T mutation affects the canonical splice acceptor site in the 15 intron and appears to contribute to modifications of collagen structure and function [38]." The link listed as reference 38 https://www.genotek.ru/diagnostic/mutations/16216/ is not working; I checked the database and could not find the mutation. Please clarify this issue and state why the variant contributes to collagen structure and function modifications.

-       lines 207-211, "The variant of c.4114A>T (p.Asn1372Tyr) have never been reported by fellow investigators and are absent in the gnomAD control sample [39]. Pathogenicity prediction in silico programs (SIFT, MT, DANN, MetalRL) consider these variants of uncertain value (VUS) and disrupting protein function. The patients' phenotypes are highly specific for the genetic etiology. The overall evidence allowed to consider such variants as of unknown clinical significance." Considering this variant is novel, the authors should better analyze it, i.e. is the aminoacid conserved among species? Does it fall in a particular domain of the protein? Can the authors make an in-3D silico prediction? All this info would allow a better evaluation of the variant's pathogenic role.

4.     In Table 1, two variants appear to be inherited. It would also be important to have the pedigree of the families to understand better the transmission and some info about the clinical status of the parents carrying the variant if they are also affected or not because if not, the authors should consider this in the Discussion as incomplete penetrance.

5.     To discuss a genotype/phenotype correlation, the authors should better describe the clinical phenotype of the reported cases, i.e. enriching the Clinical manifestations in Table 2. Adding age and laboratory analyses would also be important.

6.     In the Conclusion, the authors should justify using whole exome sequencing instead of a multi-gene panel, i.e. was the phenotype of the reported cases indistinguishable from other inherited disorders characterized by bone fragility and/or skeletal dysplasia? Was it more cost-effective?

7.     The Reference list should be revised since some references are incorrect. Some examples are references 39, 42, and 43.

Minor comments:

1.     In Table 1, please adjust the title "Table 1 Genes, types and clinical sings of osteogenesis imperfect".

2.     In Table 2, Pathogenicity, pathogenic criteria, please specify that these criteria are derived from the American College of Medical Genetics and Genomics recommendations and add the reference to the list (PMID: 25741868).

3.     In Table 2, Clinical manifestations (before medical treatment), please specify which treatment was administered.

4.     In the Results, lines 203, 204, "The c.739-2A>T mutation affects the canonical splice acceptor site in the 15 intron and appears to contribute to modifications of collagen structure and function [38]. Please specify that this variant is in the COL1A2 gene.

5.     In the Discussion, lines 250 and 251 state, "These mutations generally arise de novo [7]. The COL1A2 c.739-2A>T variant (rs72656382) was detected by us in 1 patient with OI. This variant is percent in Clinvar Database [45]." This sentence is not clear; please adjust it. It.

Author Response

Dear Reviewer,

Please see our answers to your questions in the attached file.

Best regards, A. Chernov, O. Glotov

Round 2

Reviewer 1 Report (New Reviewer)

Comments and Suggestions for Authors

The authors in the revised MS have not explained if the pathogenic variants were seen in  BMP1, CCDC134,CREB3L1, CRTAP, FKBP10, IFITM5, KDELR2, MBTPS2, MESD, P3H1, P4HB, PLOD2,PPIB, SEC24D, SERPINF1, SERPINH1, SP7, SPARC, TENT5A, TMEM38B and WNT1. The genes and their relation with diseases is mentioned the introduction. In the results if they have WES data they should indicate if they have analyzed these genes and have/have not obtained any pathogenic variant.

Also, they overlooked the first comment of analysis workflow

In the response to reviewer file the authors mentioned

" if we had found several pathogenic variants in patients, we would have conducted a “network” analysis. " at the same time they also explained "We performed whole exome analysis. All these genes were analyzed for the presence of pathogenic variants".  This is contradictory if they have several gene set data then why not network analysis. 

The author needs to be discreet when addressing reviewers comments.

Author Response

Dear Reviewers,

Please see our responses to your comments.

Please write the findings of the study briefly in the Introduction along with the rationale.

Corrections are highlighted in blue.

  1. Why did the author did not look at Variation analysis in other genes BMP1, CCDC134, CREB3L1, CRTAP, FKBP10, IFITM5, KDELR2, MBTPS2, MESD, P3H1, P4HB, PLOD2, PPIB, SEC24D, SERPINF1, SERPINH1, SP7, SPARC, TENT5A, TMEM38B and WNT1?
  2. Can the author perform a network analysis of the pathogenic variants in collagen gene with the other genes associated with the disease?

We performed whole exome analysis. All these genes were analyzed for the presence of pathogenic variants. However, pathogenic variants were found only in the COL1A1, COL1A2 genes. No pathological variants were found in the remaining genes indicated. For this reason, it is these genes (COL1A1, COL1A2) and their variants that we consider in our article. Also, since no pathogenic variants were found in other genes except (COL1A1, COL1A2), we cannot conduct a network analysis. Furthermore, the purpose of the study was primarily clinical.

  1. It would be great if authors could validate their results from data present in the publicly available data sets

We checked all our results with Russian population dataset (RuSeq), and also with databases of 1000 genomes and others.

  1. Please write the findings of the study briefly in the Introduction along with the rationale.

We briefly wrote our results in the introduction:

For example, we identified 6 genetic variants such as: pathogenic c.3505G>A (p. Gly1169Ser), c.769G>A (p.Gly257Arg), VUS c.4123G>A (p.Ala1375Thr), c.4114A>T (p.Asn1372Tyr) in the COL1A1 and likely pathogenic c.2035G>A (p.Gly679Ser), c.739-2A>T in the COL1A2 in 6 OI patients, 3 of them were 1 month of age, and two were aged up to 2 years. By this age, all patients had from 3 to 12 injuries and bone fractures, Table 2.

  1. The authors in the revised MS have not explained if the pathogenic variants were seen in  BMP1, CCDC134,CREB3L1, CRTAP, FKBP10, IFITM5, KDELR2, MBTPS2, MESD, P3H1, P4HB, PLOD2,PPIB, SEC24D, SERPINF1, SERPINH1, SP7, SPARC, TENT5A, TMEM38B and WNT1. The genes and their relation with diseases is mentioned the introduction. In the results if they have WES data they should indicate if they have analyzed these genes and have/have not obtained any pathogenic variant.

We performed whole exome analysis. All these genes were analyzed for the presence of pathogenic variants. However, pathogenic variants were found only in the COL1A1, COL1A2 genes. No pathological variants were found in the remaining genes indicated. For this reason, it is these genes (COL1A1, COL1A2) and their variants that we consider in our article. Also, since no pathogenic variants were found in other genes except (COL1A1, COL1A2), we cannot conduct a network analysis. Furthermore, the purpose of the study was primarily clinical.

Also, they overlooked the first comment of analysis workflow.

This question is not entirely clear. Please explain what needs to be done to carry out the analysis workflow?

  1. In the response to reviewer file the authors mentioned

" if we had found several pathogenic variants in patients, we would have conducted a “network” analysis. " at the same time they also explained "We performed whole exome analysis. All these genes were analyzed for the presence of pathogenic variants".  This is contradictory if they have several gene set data then why not network analysis. 

There is no contradiction here. We analyzed all these genes, but we did not find pathogenic variants in them, except for the СOL1A1, СOL1A2 genes. For this reason, we cannot conduct a network analysis. (As indicated in the first phrase “if we had found several pathogenic variants in patients, we would have conducted a “network” analysis.) The word «if» means that we did not find any variants.

This manuscript is a resubmission of an earlier submission. The following is a list of the peer review reports and author responses from that submission.

Round 1

Reviewer 1 Report

Comments and Suggestions for Authors

Major Revisions:

·         Question: A clear question is not addressed in this study. A mere report of mutations is not appropriate for publication.

·         Sequencing Quality: Please include a statistical analysis demonstrating the quality of the sequencing data employed in this study.

·         Lack of Comprehensive Analysis: The study introduces 6 cases of OI mutations without a thorough analysis. In the results section, the potential disruption of collagen structure was indicated. Please add the illustration of these predictions and conduct a more in-depth analysis.

·         Materials and Methods: Please provide the missing materials section. Please describe the methods for sequencing, and data processing and analysis, separately.

·         Conclusions: Please revise the conclusions paragraph to ensure it contains specific and relevant statements that are well-connected to the study's findings. The current version is too vague.

Minor Revisions:

·         Line 31: Please revise the sentence since the current study presents mutations without conducting a comprehensive analysis.

·         Abstract: Please modify the last sentence of the abstract to ensure it directly relates to the study.

·         Line 59: Please correct “IO” to “OI.”

·         Lines 68-69: Please provide a revised version to ensure coherence and clarity.

·         Line 101: If the statement is accurate, please include a reference to support it.

·         Line 155: Please correct the identified typo.

·         Table 1: Please redesign Table 1 for improved clarity. Please ensure the last column has sufficient space.

·         Lines 252-254: Please remove lines 252-254.

Comments on the Quality of English Language

Overall, it is fine.

Author Response

Dear Reviewer, please see attachment.

Reviewer 2 Report

Comments and Suggestions for Authors

The reviewer considers that the following should be addressed:

1. Lines 79-89. It would be easier for readers if the information was presented in a table.

2. Section 2. Please provide details on the collection, storage and processing of patient´s genetic material submitted to WES.

3. Table 1. Please provide on the caption the meanings of VUS, PM, PP3,...

4. Lines 149-169. There are several pieces of information that need bibliographic support. Please add the references concerning those paragraphs.

4.1. The same for Discussion section, for instance lines 201, 222, 226.

5.Discussion section.  Please discuss further the following:

5.1 How the molecular findings relate to clinical presentation for each of the evaluated patients.

5.2 How molecular characterization, in this specific study, will impact the prognosis and management of those patients

5.3. Please address if is cost-effective and time-effective to perform molecular diagnosis.

Comments on the Quality of English Language

Thorough revision is required.

Author Response

Dear Reviewer, please see attachment.

Round 2

Reviewer 1 Report

Comments and Suggestions for Authors

The authors did not sufficiently respond to the comments. The response letter was not written to show the revision to each of the comments.

Comments on the Quality of English Language

Fine.

Reviewer 2 Report

Comments and Suggestions for Authors

The reviewer acknowledges the author´s revisions.

Comments on the Quality of English Language

English revision is required.